# Impact of Depression and Anxiety on Dimensions of Health-Related Quality of Life in Subjects with Parkinson’s Disease Enrolled in an Association of Patients

**DOI:** 10.3390/brainsci11060771

**Published:** 2021-06-10

**Authors:** Fany Chuquilín-Arista, Tania Álvarez-Avellón, Manuel Menéndez-González

**Affiliations:** 1Community and Family Medicine, Health Area VII—Asturias, Plaza de los Sindicatos Mineros 3, 33600 Mieres, Spain; 2Health Science Research, Doctoral School, University of Valladolid, Calle Real de Burgos s/n, 47011 Valladolid, Spain; 3Department of Psychology, Universidad de Oviedo, Plaza Feijoo s/n, 33003 Oviedo, Spain; tania.avellon@gmail.com; 4Department of Neurology, Hospital Universitario Central de Asturias (HUCA), Avenida Roma s/n, 33011 Oviedo, Spain; menendezgmanuel@uniovi.es; 5Instituto de Salud del Principado de Asturias (ISPA), Avenida del Hospital Universitario s/n, 33011 Oviedo, Spain; 6Department of Medicine, Universidad de Oviedo, Calle Julián Clavería 6, 33006 Oviedo, Spain

**Keywords:** Parkinson’s disease, depression, anxiety, quality of life, nonmotor symptoms

## Abstract

Parkinson’s disease (PD) is a complex disorder characterized by a wide spectrum of symptoms. Depression and anxiety are common manifestations in PD and may be determinants of health-related quality of life (HRQoL). The objective of this study is to determine the association of depression and anxiety with the dimensions of HRQoL in subjects with PD enrolled in an association of patients. Ninety-five community-based patients with PD diagnosis at different disease stages were studied. HRQoL was assessed using the Parkinson’s Disease Questionnaire (PDQ-39); depression and anxiety were assessed using the Beck Depression Inventory (BDI-II) and the State-Trait Anxiety Inventory (STAI), respectively. Our results showed that depression and anxiety were negatively associated with HRQoL measured by PDSI. Higher motor dysfunction measured by Hoehn and Yahr (H&Y) staging was also associated with worse HRQoL. Depression was the most influential variable in the model. All PDQ-39 dimensions except social support and bodily discomfort were associated with depression. Anxiety was associated with the emotional well-being and bodily discomfort dimensions. These results suggest that physicians should pay attention to the presence of psychiatric symptoms and treat them appropriately.

## 1. Introduction

Quality of life (QoL) is defined by the World Health Organization (WHO) as “individuals’ perception of their position in life in the context of the culture and value systems in which they live and in relation to their goals, expectations, standards, and concerns” [1]. In the clinical setting, there is a general agreement to focus on health-related quality of life (HRQoL). HRQoL is defined as “the perception and evaluation by patients themselves of the impact caused on their life by the disease and its consequences” [2].

Parkinson’s disease (PD) is a neurodegenerative disease found mostly among older adults. It is estimated that PD affects 1–2% of people over 65 years of age [3]. The disease involves prominent motor symptoms but also causes neuropsychiatric symptoms such as depression and anxiety. These neuropsychiatric symptoms are frequently overlooked and/or unreported [4] and might have a great impact on the patient’s QoL and on the different dimensions of HRQoL [5].

Although several previous studies have studied the prevalence of anxiety and depression in PD and their impact on QoL [6,7,8,9,10,11,12,13,14,15,16], the populations of these studies were heterogeneous from an ecosocial point of view, since both patients institutionalized in geriatric residencies and those living at home were included. Those patients may or may not be attending activities at associations of patients. All these factors may represent important biases when assessing anxiety, depression, and QoL. Moreover, most previous studies assessed the overall QoL or HRQoL, while the analysis of the different dimensions was overlooked.

The aim of this study is to determine the association of depression and anxiety with HRQoL and its specific dimensions in an ecosocially homogeneous cohort of PD patients living at home who attend activities at an association of patients.

## 2. Materials and Methods

### 2.1. Study Design and Patient Population

This is an observational, descriptive, community-based, survey-type study. It was approved by the Ethical Committee of Investigation of the Principality of Asturias. The design of this study complies with the checklist of items of the STROBE Statement Checklist for reports of cross-sectional studies.

The study was done on a cohort of subjects diagnosed with PD according to UK Brain Bank criteria. One hundred and four subjects participated in the study. After screening, the final study population was composed of 95 subjects. All of them were recruited from an association of patients (the “Asturias Parkinson Association”) during March 2017. Informed consent was obtained from all participants. The Mini-Mental State Examination (MMSE) was administered as a brief screen for dementia. Subjects younger than 18 years of age, with a score of less than 12 in the MMSE, unable to complete self-administered questionnaires, with difficulty writing and reading, with history of psychosis, and/or with an associated disease in terminal phase or receiving palliative treatment were excluded. Informed consent was obtained for each subject. Subjects were questioned about demographic characteristics and medical history. The Hoenh and Yahr (H&Y) scale was also included. Later, patients completed the Beck-II, STAI, and PDQ-39 tests.

### 2.2. Study Variables

We analyzed sociodemographic variables (age, gender, civil status, education level, labor sector) and clinical variables (time from diagnosis of PD, age at onset of PD, severity of PD, cognitive status, PD and other disease treatment, history of anxiety and depression, quality of life). The assessment of the severity of PD was made using the H&Y staging. The cognitive status was divided into three conditions based on the result of the MMSE score adjusted by age and educational level: normal, ≥27; suspected impairment, ≥23 and <27; and cognitive impairment, <23 [17]. These cut-off points were used to increase the sensitivity of the test.

The Beck Depression Inventory II (BDI-II) was administered to assess depression symptomatology. This test consists of 21 questions and each question is scored 0–3. The depression severity was measured using the Beck classification of 1998: without depression (0–13), mild depression (14–19), moderate depression (20–28), and severe depression (29–63) [18].

The Spielberg Trait Anxiety Inventory (STAI-T) was administered to assess anxiety. STAI-T is a 20-item self-report instrument designed to assess anxiety as a latent trait (Anxiety Trait). Each item is graded from 0 to 3 depending on the anxiety intensity. The values obtained were converted to centiles, according to the Spanish version of the STAI [19]. We considered the presence of anxiety at STAI-T > p50 (50th percentile). Also, due to the high variation in results, we divided the anxiety into low (STAI-T ≤ p25), mild–low (p25 < STAI-T ≤ p50), medium–high (p50 < STAI-T ≤ p75), and very high (STAI-T > p75).

The Parkinson’s disease Quality of Life Questionnaire (PDQ-39) was administered to assess HRQoL [20]. This is a disease-specific, 39-item questionnaire QoL measure. It contains 8 dimensions (mobility, activities of daily living, emotional well-being, stigma, social support, cognition, communication, and bodily discomfort). Each question is scored from 1 to 4 according to the frequency of the mentioned symptoms and is expressed as a percentage. Subsequently, the results were used to obtain the Parkinson Disease Summary Index (PDSI). Index values range from 0 to 100, and a higher index value indicates higher health status.

All three tests showed a high reliability (Cronbach’s alpha > 0.7). The alpha values were 0.857 for BDI-II, 0.861 for STAI-T, and higher than 0.7 in all dimensions of PDQ-39 (mobility: 0.878, activities of daily living: 0.885, emotional well-being: 0.856, stigma: 0.703, social support: 0.701, cognition: 0.702, communication: 0.808, and bodily discomfort: 0.708).

### 2.3. Statistical Analysis

Data were processed using a Microsoft Excel database that was subsequently imported to R (R Development Core Team), version 3.4.4, for analysis. Qualitative variables were expressed as absolute and relative frequencies; the latter were measured as percentages. The relationships between qualitative variables were assessed using Pearson’s chi-squared test or Fisher’s exact test. The differences in quantitative variables between two groups were studied using Student’s *t*-test or the Wilcoxon test depending on whether or not the expected frequency assumption was verified. To compare 3 or more groups, the test used was the ANOVA or the Kruskal–Wallis test. Multiple linear regression analyses were performed for each of the dimensions of the PDQ-39, as well as for PDSI. We adjusted for depression, anxiety, and variables that were statistically significant in previous analyses. We did not introduce anxiety in the social support multivariate model because it has strong collinearity with it. A *p* value of less than 0.05 was used to assess the statistical significance of interaction terms. Finally, squared partial correlations were made to see the contribution of the main variables considered in the model of PDSI.

## 3. Results

### 3.1. Description of the Population

In total, 95 subjects were studied: 56 men (59%) and 39 women (41%). The overall mean age was 70.78 ± 8 (70.73 ± 8.02 for men and 71.28 ± 8.05 for women), 70.53% were married (92.86% of men and 38.46% of women), and 29.47% were single, divorced, or a widow(er) at the moment of the study.

Concerning education level, 5.26% were uneducated, 42.11% had a primary education, 28.42% had secondary education, and 24.21% had a higher education level. The main labor sector was the service sector (48.42%).

The mean age at onset of the disease was 62.63 ± 10.36 years and the median of the disease duration was 8.15 ± 6.35 years. More than half (66.32%) had no family history of PD.

The most frequent H&Y stages were 1 and 3 (26.32% and 25.26%, respectively).

Cardiovascular disease was the most frequent comorbidity (41.05%). The MMSE showed cognitive impairment in 8.42% and suspected impairment in 7.37%. Depression had already been diagnosed in 20% of subjects, 9.47% had a diagnosis of anxiety, and 1.1% had both of these. Furthermore, 25.26% had been taking antidepressants and 38.95% had been taking anxiolytics (benzodiazepines and benzodiazepine analogues).

Table 1 shows the main characteristics of the studied population.

### 3.2. Depression and Anxiety

The mean depression score on the BDI-II was 13.29 (±8.51) and the mean anxiety score on the STAI-T was 26.02: 24.86 (±8.83) for men and 27.69 (±10.05) for women (74th and 63rd percentiles, respectively).

Depression was present in 32.63% of subjects (25% of men and 43.59% of women). Furthermore, 7.37% had severe depression.

Anxiety (STAI-T > p50) was present in 68.42% of subjects, while 42.11% had very high anxiety levels (STAI-T > p75), and it was more frequent in males. Comorbidity of depression and anxiety was present in 31.58%.

Table 2 shows the occurrence of depression and anxiety in the studied population.

### 3.3. Health-Related Quality of Life (HRQoL)

The mean score of the PDSI was 26.47 (±13.64). Bodily discomfort had the highest score with a mean of 36.75 (±23.17). Stigma had the lowest score with a mean of 10.32 (±13.63). When analyzing the results by sex, it was found that the most affected dimensions were mobility in men and bodily discomfort in women. The scores of the PDQ-39 dimensions and PDSI are presented in Table 3.

### 3.4. Bivariate Analysis of the PDQ-39 Dimensions and PDSI

We analyzed the data of all our 95 patients to find the associations of the PDSI and PDQ-39 dimensions with sociodemographic and clinical characteristics and with BDI-II and STAI-T.

The bivariate analysis between PDSI and sociodemographic characteristics did not show significant differences. There was an association between PDSI and disease duration (*p* = 0.039), H&Y scale (*p* = 0.001), subjects that had already been diagnosed with depression (*p* = 0.026), and those who were already taking antidepressants (*p* = 0.017). There was a strong association of PDSI with patients with depressive and anxious symptoms measured via the BDI-II and STAI-T tests (*p* < 0.001).

In analyzing the dimensions of the PDQ-39, an association was found between the female sex and worse emotional well-being (*p* = 0.003), cognition (*p* = 0.03), and bodily discomfort (*p* = 0.002). Male sex and married subjects were associated with worse results in the communication dimension (*p* = 0.013). In the labor sector, work as a housewife was associated with further deterioration of the cognition dimension. Subjects with pulmonary disease performed worse in the activities of daily living dimension (0.026), and those with cardiovascular disease performed worse in the mobility dimension (*p* = 0.023).

PD duration was negatively associated with activities of daily living (*p* = 0.039), and higher H&Y stage was found to be significantly associated with reductions in the mobility (*p* < 0.001), activities of daily living (*p* = 0.001), emotional well-being (*p* = 0.039), and cognition (*p* = 0.017) dimensions.

There was a negative statistical association between worse results in the bodily discomfort dimension and subjects without family history of PD (*p* = 0.028) and those with suspected cognitive impairment (*p* = 0.02). Worse results in the cognition dimension were also observed in subjects who had been taking antidepressants (*p* = 0.015) and had already been diagnosed with depression (*p* = 0.003). These variables were also related to worse emotional well-being (*p* ≤ 0.001 and *p* = 0.003, respectively).

The dimension of social support was negatively related to age of onset of ≥50 years (*p* = 0.013) and to people without an anxiety diagnosis at the time of the study (*p* = 0.004).

All dimensions except social support were related to depressive and anxious symptoms (*p* < 0.05).

### 3.5. Multiple Linear Regression Analysis

Table 4 shows the multiple linear regression analyses with the influences on the PDSI and PDQ-39 dimensions. The significant model (adjusted R^2^: 0.4908) revealed significant influences of motor manifestations of the disease (4.0 H&Y stage; *p* = 0.013) and patients with depression (*p* < 0.001) and anxiety (*p* < 0.015) on HRQoL measured by PDSI.

When analyzing the dimensions of the PDQ-39, we found that all dimensions except for social support and bodily discomfort had a strong statistical association with the presence of depression (*p* < 0.005). Anxiety was also an independent determinant of emotional well-being (*p* < 0.001) and bodily discomfort (*p* = 0.014).

Being male was positively associated with the emotional well-being (*p* = 0.006), cognition (*p* = 0.003), and bodily discomfort (*p* = 0.001) dimensions and negatively associated with communication (*p* = 0.001).

Squared partial correlations showed that the most influential variable in the model was depression (23.30%), followed by the H&Y stage (23.02%) and anxiety (8.24%).

There was association between H&Y stages and the dimensions of mobility, activities of daily living, emotional well-being, and cognition (*p* < 0.05).

The social support dimension was associated with the age of onset of PD of ≥50 years (*p* = 0.030).

Among chronic pathologies, cardiovascular disease and pulmonary disease were associated with worse mobility and activities of daily living, respectively.

Finally, primary education level and the use of more than one antiparkinsonian drug were positively associated with the activities of daily living dimension.

## 4. Discussion

The main finding of this study is that neuropsychiatric symptoms (such as depression and anxiety) have a great influence on global HRQoL in patients with PD. Considered together, depression and anxiety explained about 31% of the variance of HRQoL. Moreover, depression was the main determinant of HRQoL, even more than motor impairment, and was negatively associated with most HRQoL dimensions.

The contribution of depression to impaired HRQoL in PD [6,7,8,9,10,11,12] and most HRQoL dimensions [10,21,22] is a consistent finding in the literature, even in studies using different methodologies. Indeed, a German study that grouped the PDQ-39 dimensions into three components (physical functioning, cognition, and socioemotional HRQoL) found similar results [23]. This means that subjects with better mood status have a better perception of their quality of life not only in the psychosocial but also in the physical functioning domains.

Anxiety was another significant determinant of HRQoL and had a high prevalence in our community-based sample. This is consistent with other studies [9,10,24,25]. The mechanism of this association is unclear. We know that anxiety symptoms are more prevalent in PD patients than in the general population or in subjects with other chronic diseases [26]. Besides anxiety itself, an anxious personality has also been revealed as a risk factor for PD much later in life [27]. The association between HRQoL dimensions and anxiety (the well-being and bodily discomfort dimensions) could be explained by physical discomfort increasing when the patient is concentrating or feeling anxious, deteriorating their sense of well-being.

Regarding demographics, we found no association between age, civil status, or family history and global HRQoL measured by PDSI and any PDQ-39 dimensions in multivariate analysis. The association between gender and HRQoL and its dimensions differed in previous studies [8,13,14,15,16]. Ophey et al. [23] studied 245 patients in Germany and found that female gender was a negative predictor for physical functioning and socioemotional HRQoL, whereas male gender was a negative predictor for cognition QoL. Moreover, a study from Egypt found that those with female gender and low socioeconomic status were more vulnerable to depression, while anxiety was recorded more in young ages. Both depression and anxiety cause impairment of QoL in a similar manner [16]. We found the worst quality of life in men in the communication dimension and lower QoL in women in the emotional well-being, bodily discomfort, and cognition dimensions. There was no influence of gender on global quality of life. The different results across various studies could be explained by the complexity of HRQoL. Moreover, the perception of quality of life may be different in men and women, influenced by hormonal, cultural, and social aspects. In this context, Moore et al. [28] examined the correlation between gender identity and quality of life; they found that men and women with a high number of both feminine and masculine characteristics (androgynous PD patients) cope better in terms of quality of life, especially androgynous women.

Regarding education, some studies indicate that it has a protective effect on cognitive functions and may influence quality of life [7,29]. We found that people with a primary education level had less risk of deterioration in the activities of daily living dimension but found no association of level of education with the cognition dimension or global quality of life.

With regard to influences of other pathologies on HRQoL, we found that history of cardiovascular disease was related to worse performance in the mobility dimension, and pulmonary disease was related to worse results for activities of daily living. This could be because people with those chronic pathologies have even more difficulty in movement and in doing activities that they used to do. However, those pathologies had no impact on global HRQoL.

Some studies correlated cognitive impairment to reduced HRQoL [6,9,30]. We found an association between suspected cognitive impairment and the bodily discomfort dimension but, surprisingly, we found no association with global quality of life or the cognition dimension. This might be due to the loss of insight in PD with mild cognitive impairment. Orfei et al. [31] showed that anosognosia for non-motor symptoms is frequent in PD patients with mild cognitive impairment; therefore, many patients might not be aware of their non-motor status. Another study showed that PD patients with cognitive complaints had more anxiety and depression [32]. Surprisingly, that study also showed that PD patients without cognitive complaints had poorer cognitive performance compared to controls but were similar to the PD patients with complaints. These results reinforce the idea that cognitive complaints in PD patients could point to mood disorders instead of real cognitive impairment, while the absence of cognitive complaints could be related to anosognosia.

In bivariate analysis, we found worse results in the cognition dimension in subjects who had been taking antidepressants and had already been diagnosed with depression. These variables were also related to a worse emotional well-being dimension and were not statistically significant in multivariate analysis. This finding does not necessarily mean that antidepressants worsen cognitive functions, but most probably it is the mood disorder that conditions cognition. Indeed, the effect of antidepressants on cognitive function seems to be either positive or neutral. A metanalysis of the effects of antidepressants on cognitive functioning in depressed and non-depressed people showed that overall, antidepressants have a modest, positive effect on divided attention, executive function, immediate memory, processing speed, recent memory, and sustained attention for depressed participants [33]. A randomized controlled trial in PD found that treatment responders did not exhibit larger gains in cognition than non-responders [34]. An open label trial found that Citalopram did not significantly change cognition of PD patients from baseline to endpoint, while depression improved significantly and was associated with significant improvements in anxiety symptoms and functional impairment [35].

Contrary to what we expected, we found that subjects that received more than one treatment showed a positive association with the HRQOL activities of daily living dimension. This could be because they have greater control of their symptoms, and this would allow them to better perform the activities of daily life. However, it is also important to remember the placebo effect that is inherent to most therapies, including antiparkinsonian drugs. In any case, to reach more valid conclusions, other variables that we did not consider would need to be included, such as the levodopa equivalent dose, number of pills per day, advanced therapies such as deep brain stimulation, and motor and non-motor scores, in order to correlate them all together with HRQoL.

In contrast with other studies that found that early onset resulted in a poorer HRQoL and worse scores on the stigma dimension [36,37], we did not find such associations. We found an association between age of onset at 50 or above and worse results in the social support dimension. This could be explained by the fact that the average age of patients with early onset PD was 10 years younger than that of other patients, and it is expected that younger patients have younger and healthier partners who can provide more help and support—according to a Spanish sociodemographic study, most couples have an age difference of ≤4 years [38].

In agreement with other studies, we found that H&Y stage was a significant determinant for the HRQoL [6,9,30,39,40]. It was also related to the mobility, activities of daily living, emotional well-being, and cognition dimensions. In the mobility dimension, having difficulty doing pleasurable leisure activities negatively affected HRQOL. These findings were also reported in other studies [41,42]. Moreover, increasing physical impairment affects activities of daily living by increasing the time required to perform them or dependence on others, reducing personal satisfaction and adversely affecting the emotional well-being dimension. Problems related to concentration, poor memory, and bodily pain also increased with disease severity and were negatively associated with health-related quality of life.

Altogether, our study contributes to a better understanding of anxiety and depression in PD and their impact on HRQoL in community-based populations. There are still some important limitations, including: (1) the sample size and the selection of subjects from an association of patients, because they could have better social support than others; (2) the use of a brief self-report measure of anxiety and depression, without a direct assessment of the mental health of subjects, which does not allow a formal diagnosis; and (3) failing to take into consideration more detailed data about treatments, including levodopa equivalent dose, number of pills per day, and advanced therapies. Moreover, a prospective study would also be helpful to better capture changes in the HRQL.

## 5. Conclusions

In summary, depression, anxiety, and a more advanced stage of PD had a negative association with HRQoL in PD patients enrolled in an association of patients. Depression affects most of the HRQoL dimensions, while anxiety mainly affects the emotional well-being and bodily discomfort dimensions. The findings of the present study suggest that, besides treatment of motor symptoms of PD, an assessment and management of depression and anxiety would contribute to improving the HRQoL of patients with PD. We believe that a multidisciplinary (neurologist, psychiatrist, psychologist, rehabilitation therapists, and other health care professionals) approach to treatment should be assumed.

## Figures and Tables

**Table 1 brainsci-11-00771-t001:** Characteristics of the population.

Characteristic	Sex						*p*-Value
	Male		Female		Total		
	N	%	N	%	N	%	
*Civil status*							
Single	1	4.21	3	7.69	4	4.21	
Married	52	92.86	15	38.46	67	70.53	<0.001
Widow(er)/divorced	3	5.36	21	53.85	24	25.26	
*Education level*							
None	4	7.14	1	2.56	5	5.26	0.157
Primary	19	33.93	21	53.85	40	42.11	
Secondary	16	28.57	11	28.21	27	28.42	
Higher education	17	30.36	6	15.38	23	24.21	
*Labor sector*							
Primary	11	19.64	0	0.00	11	11.58	
Industrial	19	33.93	1	2.57	20	21.05	<0.001
Service	26	46.43	20	51.28	46	48.42	
Housewife	0	0.00	18	46.15	18	18.95	
*Family history of PD*							
No	37	66.07	26	66.67	63	66.32	0.919
Yes	17	30.36	11	28.21	28	29.47	
Do not know	2	3.57	2	5.13	4	4.21	
*Age of onset*							
<50 years	5	8.93	5	12.82	10	10.53	0.736
≥50 years	51	91.07	34	87.18	85	89.47	
*Hoehn and Yahr stage*							
0	2	3.57	1	2.56	3	3.16	
1.0	14	25.00	11	28.21	25	26.32	
1.5	5	8.93	4	10.26	9	9.47	
2.0	4	7.14	3	7.69	7	7.37	0.967
2.5	10	17.86	4	10.26	14	14.74	
3.0	13	23.21	11	28.21	24	25.26	
4.0	8	14.29	5	12.82	13	13.68	
5.0	0	0.00	0	0.00	0	0.00	
*Cognitive status*							
Normal	48	85.71	32	82.05	80	84.21	
Suspected impairment	5	8.93	2	5.13	7	7.37	0.366
Cognitive impairment	3	5.36	5	12.82	8	8.42	
*Organic diseases*							
Cardiovascular	22	39.29	17	43.59	39	41.05	0.675
Pulmonary	9	16.07	0	0.00	9	9.57	0.009
Digestive	3	5.36	1	2.56	4	4.21	0.505
Oncological	2	3.57	2	5.13	4	4.21	0.710
Others	38	67.86	30	76.92	68	71.58	0.335
*History of mental disorders*							
Anxiety	5	8.93	4	10.26	9	9.47	0.828
Depression	7	12.50	12	30.77	19	20.00	0.029
Others	16	28.57	4	10.26	20	21.05	0.031

**Table 2 brainsci-11-00771-t002:** Occurrence of depression and anxiety.

Depression and Anxiety	Sex						*p*-Value
	Male		Female		Total		
	N	%	N	%	N	%	
*BDI-II*							
Without depression	42	75.00	22	56.41	64	67.37	0.293
Mild depression	4	7.14	4	10.26	8	8.42	
Moderate depression	7	12.50	9	23.07	16	16.84	
Severe depression	3	5.36	4	10.26	7	7.37	
*STAI-T*							
Low	9	16.07	6	15.38	15	15.79	
Mild–low	8	14.28	7	17.95	15	15.79	
Medium–high	10	17.86	15	38.46	25	26.31	0.074
Very high	29	51.79	11	28.21	40	42.11	

The Beck Depression Inventory II: BDI-II, without depression (BDI-II: 0–13), mild depression (BDI-II: 14–19), moderate depression (BDI-II: 20–28), and severe depression (BDI-II: 29–63). STAI-T: The Spielberg Trait Anxiety Inventory, low (STAI-T ≤ p25), mild–low (p25 < STAI-T ≤ p50), medium–high (p50 < STAI-T ≤ p75), and very high (STAI-T > p75). Anxiety was considered to be present at STAI-T > p50.

**Table 3 brainsci-11-00771-t003:** Dimensions of the PDQ-39 and PDSI in the study population.

HRQoL	Sex						*p*-Value
	Male		Female		Total		
	Mean (Percentile)	SD	Mean (Percentile)	SD	Mean (Percentile)	SD	
*PDQ-39 Dimensions*							
Mobility	31.21	21.17	35.26	26.80	32.87	23.59	0.414
Activities of daily living	29.61	23.91	24.57	27.98	27.54	25.63	0.349
Emotional well-being	25.52	17.71	38.03	22.90	30.66	20.83	0.003
Stigma	8.37	10.27	13.14	17.13	10.32	13.63	0.125
Social support	21.88	23.81	17.09	21.54	19.91	22.91	0.320
Cognition	28.24	18.62	41.03	22.25	33.49	21.05	0.003
Communication	24.85	23.00	13.46	19.55	20.18	22.27	0.013
Bodily discomfort	30.80	22.75	45.30	21.27	36.75	23.17	0.002
*PDSI*	25.06	12.94	28.49	14.53	26.47	13.64	0.231

PDQ-39: The 39-item Parkinson’s Disease Questionnaire; PDSI: Parkinson’s Disease Summary Index; HRQoL: Health-related quality of life; SD: Standard deviation. The PDQ39 and PDSI are expressed in percentiles. For each patient, the percentile and the corresponding mean values were computed.

**Table 4 brainsci-11-00771-t004:** Multivariate linear regression analysis of the dimensions of the PDQ-39 and PDSI. Significant values: * *p* < 0.05, ** *p* ≤ 0.01, *** *p* ≤ 0.001.

Characteristic	Mobility	ADL	Emotional Well-Being	Stigma	Social Support	Cognition	COMM	Bodily Discomfort	PDSI
Male sex	-	-	−9.501 **	-	-	−14.690 **	14.708 ***	−15.935 ***	-
Education									
Prim educ	-	−19.916 *	−2.724	-	-	−10.254	-	-	−5.500
Sec educ	-	−10.033	6.384	-	-	−3.862	-	-	−0.308
High educ	-	−19.108	2.349	-	-	−13.301	-	-	−7.136
Labor sector									
Industrial	-	-	-	-	-	4.320	-	-	-
Service	-	-	-	-	-	−9.198	-	-	-
Housewife	-	-	-	-	-	−3.394	-	-	-
FH of PD	-	-	-	-	-	-	-	−9.209	-
Age of onset (≥50 yrs)	-	-	-	-	16.462 *	-	-	-	-
H&Y stage									
Stage 1.0	2.191	12.799	24.395 **	-	-	10.547	-	-	2.843
Stage 1.5	15.799	16.987	31.801 ***	-	-	20.986	-	-	9.914
Stage 2.0	−5.802	8.780	21.462 *	-	-	1.416	-	-	−2.190
Stage 2.5	6.316	18.889	35.049 ***	-	-	24.262 *	-	-	10.307
Stage 3.0	8.994	27.755 *	22.300 *	-	-	11.798	-	-	6.782
Stage 4.0	35.241 **	43.954 ***	31.939 ***	-	-	23.514 *	-	-	16.218 *
CS (MMSE)									
Suspec impair	-	-	-	-	-	-	-	23.930 **	-
Cogn impair	-	-	-	-	-	-	-	2.019	-
>1 treatment	-	−12.702 *	-	6.355	-	-	-	-	-
Other diseases									
CVD	8.399 *	-	-	-	-	-	-	-	-
PD	-	21.810 **	-	-	-	-	-	-	-
AD treatment	-	-	6.169	-	-	-	-	-	-
BDI-II	12.204 **	15.856 **	16.812 ***	8.318 **	6.946	14.234 ***	18.374 ***	−0.926	12.423 ***
STAI-T	7.976	2.791	14.011 ***	4.851	-	5.622	3.266	13.017 *	6.019 *
Adjusted R^2^	0.3982	0.4005	0.5500	0.1307	0.0506	0.4320	0.2120	0.2524	0.4908

PDQ-39: The 39-item Parkinson’s Disease Questionnaire; PDSI: Parkinson’s Disease Summary Index; ADL: Activities of daily living; COMM: Communication; Prim educ: Primary education; Sec educ: Secondary education; High educ: Higher education; FH of PD: Family history of Parkinson’s disease; H&Y stage: Hoehn and Yoehn stage; CS: Cognitive status; MMSE: Mini-Mental State Examination; Suspec Impair: Suspected impairment; Cogn impair: Cognitive impairment; CV disease: Cardiovascular disease; PD disease: Pulmonary disease; AD: Antidepressant; BDI-II: The Beck Depression Inventory-II; STAI-T: The State-Trait Anxiety Inventory-Trait scale.

## Data Availability

The data presented in this study are available on request from the corresponding author.

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
