# Peer review of "Impact of Depression and Anxiety on Dimensions of Health-Related Quality of Life in Subjects with Parkinson’s Disease Enrolled in an Association of Patients"

_brainsci, 2021, doi:10.3390/brainsci11060771_

Round 1
Reviewer 1 Report
Thank you for the opportunity to review this manuscript. In this study Chuquilin-Arista and al investigated the association of depression and anxiety with the health-related quality of life (HQoL) in Parkinson’s disease (PD), and demonstrated a negative association. However, these findings are by no means novel and have already been shown in several studies1-3and I thus do not think that they add to the existing knowledge or meet the standards for publication in a high impact journal such as Brain Sciences.
1Quelhas R, Costa M. Anxiety, depression, and quality of life in Parkinson's disease. J Neuropsychiatry Clin Neurosci. 2009 Fall;21(4):413-9. doi: 10.1176/jnp.2009.21.4.413. PMID: 19996250.
2Hanna KK, Cronin-Golomb A. Impact of anxiety on quality of life in Parkinson's disease. Parkinsons Dis. 2012;2012:640707. doi:10.1155/2012/640707
3Khedr, E.M., Abdelrahman, A.A., Elserogy, Y. et al. Depression and anxiety among patients with Parkinson’s disease: frequency, risk factors, and impact on quality of life. Egypt J Neurol Psychiatry Neurosurg 56, 116 (2020). https://doi.org/10.1186/s41983-020-00253-5
Reviewer 2 Report
The manuscript (brainsci-1227820) titled „The Association of Depression and Anxiety with 2 Health-Related Quality of Life in Patients with Parkinson’s 3 Disease” by Chuquilín-Arista et al. is a clinical study dealing with the problem of depression and anxiety coexisting in the course of Parkinson's disease, i.e. symptoms that seem to be closely related to the quality of life of patients. Health-related quality of life (HRQoL) was evaluated by PD patients using the Parkinson’s Disease Questionnaire (PDQ-39), while depression and anxiety were assessed using the Beck-Depression Inventory (BDI-II) and the State-Trait Anxiety Inventory (STAI), respectively.
While the presented data sounds interesting, som of them have been previously published in J Geriatr Psychiatry Neurol. 2020, 33(4), 207-213, so it is hard to find a novelty in this study.
Moreover, several issues should be clarified.
In section 2.1 it is stated that the study was conducted on a group of 95 diagnosed parkinsonian patients, while in Table 1, characterizing the study population for "Other organic disease" it is stated that 124 patients were examined. Similarly, when characterizing the study population for "mental disorder", it was reported that a total of 48 patients were studied: 9 with anxiety, 19 with depression and 20 with other disorders. What about the rest of this population. So, how was the percentage of those in whom depression or anxiety coexisting in this Parkinson's population was calculated?
In Table 2, the mean and standard deviation are given for each tested parameter, but the value n (number) for each parameter is not given.
It is not clear on what basis the authors suggest that administration of antidepressants worsens cognitive functions.
In the studied group of patients (Table 1), only 8 patients had pronounced cognitive impairment, but there is no information about the type of antidepressant drugs administered and their effect on cognitive functions.
In general, Multivariate linear regression analysis presented in table 3 is unclear and difficult to interpret.
Reviewer 3 Report
The authors investigated the association between depression and anxiety with health-related quality of life in a sample including 95 patients with a diagnosis of Parkinson's disease in different stages of the disease. The topic is of high interest as psychiatric symptoms in patients with neurodegenerative disease can greatly contribute to the toll exerted by the disorder and require a comprehensive treatment.
The authors collected an extensive set of variables and the statistical analysis appears to be sound. The article is clear and well written. I have a few observations and comments to improve some parts.
- The introduction is quite short and might benefit of an extension, for istance expanding the part relative to psychiatric symptoms in patients with PD, reporting more specific data on their prevalence, characteristics and impact on the course of the disorder.
- In Table 1, I would define the two groups either "men" and "women" or "male" and "female" rather than "men" and "female".
- Table 1 should also be revised as some rows are not entirely visible and some data are aligned in a different way compared to others. Also, I would suggest to replace "Others organic disease" with "Other organic diseases"
- In the Discussion, please revise the sentence at page 8, lines 265-268 as it is not clear.
Round 2
Reviewer 1 Report
My comments have been addressed.
Author Response
We have now revised the writing throughout the manuscript.
Reviewer 2 Report
I am not satisfied with the authors' responses to some of my comments.
As shown in Table 1, of the total diagnosed population of 95 PD patients, 48 but not 45 (as reported in the authors' answer), suffered from mental disorders (anxiety - 9, depression -19, others 20) while 47 had no diagnoses of mental disorder. In addition, I did not receive information on how the % of patients with depression or anxiety was calculated. Taking into account data presented in the Table 1, 19 depressed patients (7 male and 12 female) out of 95 diagnosed PD patients (19/95) constitute 20% of the studied population, but not 32,63% as stated in line 163 on page 5. Similarly the % of depressed male (7/95) and female (12/95) is 7.3% and 12.6 %,respectively, but not 25% and 43,59 % as stated in the text (page 5, line 163). As for patients with anxiety (9/95), they constitute 9.5% of the studied population of patients with Parkinson's disease. So what is meant by the information that 68.42% of patients have anxiety symptoms (page 5, line 165). What is the point of showing the percentage of patients with a certain degree of anxiety in a group of patients with anxiety of only 9 people?
The example of statistical analysis presented above calls into question the statistical analysis provided in the remainder of the paper.
It is still not clear on what basis the authors suggest that administration of antidepressants worsens cognitive functions. This claim is unfounded without providing concrete evidence. Some antidepressants may have an effect on the cholinergic system, but as shown by recent studies, blockade of the M4 muscarinic receptors in the striatum may also have a beneficial anti-dyskinetic effect, which may indirectly affect the mental state and quality of life. Moreover, L-DOPA itself, commonly used in PD therapy, may have a psychotic effect. Hence, without an analysis of the antiparkinsonian and antidepressant drugs used in the studied population of patients, the conclusion of the deterioration of cognitive functions is not justified.
